# Generalized Tensor Models for Recurrent Neural Networks

**Valentin Khrulkov**[1], **Oleksii Hrinchuk**[1,2] **& Ivan Oseledets**[1,3]
{valentin.khrulkov, oleksii.hrinchuk, i.oseledets}@skoltech.ru
[1]Skolkovo Institute of Science and Technology, Moscow, Russia
[2]Moscow Institute of Physics and Technology, Moscow, Russia
[3]Institute of Numerical Mathematics, Russian Academy of Sciences, Moscow, Russia

## Abstract

Recurrent Neural Networks (RNNs) are very successful at solving challenging problems with sequential data. However, this observed efficiency is not yet entirely explained by theory. It is known that a certain class of multiplicative RNNs enjoys the property of depth efficiency — a shallow network of exponentially large width is necessary to realize the same score function as computed by such an RNN. Such networks, however, are not very often applied to real life tasks. In this work, we attempt to reduce the gap between theory and practice by extending the theoretical analysis to RNNs which employ various nonlinearities, such as Rectified Linear Unit (ReLU), and show that they also benefit from properties of universality and depth efficiency. Our theoretical results are verified by a series of extensive computational experiments.

## 1 Introduction

Recurrent Neural Networks are firmly established to be one of the best deep learning techniques when the task at hand requires processing sequential data, such as text, audio, or video (Graves et al., 2013; Mikolov et al., 2011; Gers et al., 1999). The ability of these neural networks to efficiently represent a rich class of functions with a relatively small number of parameters is often referred to as *depth efficiency*, and the theory behind this phenomenon is not yet fully understood. A recent line of work (Cohen & Shashua, 2016; Cohen et al., 2016; Khrulkov et al., 2018; Cohen et al., 2018) focuses on comparing various deep learning architectures in terms of their *expressive power*.

It was shown in (Cohen et al., 2016) that ConvNets with product pooling are *exponentially* more expressive than shallow networks, that is there exist functions realized by ConvNets which require an exponentially large number of parameters in order to be realized by shallow nets. A similar result also holds for RNNs with multiplicative recurrent cells (Khrulkov et al., 2018). We aim to extend this analysis to RNNs with rectifier nonlinearities which are often used in practice. The main challenge of such analysis is that the tools used for analyzing multiplicative networks, namely, properties of standard *tensor decompositions* and ideas from algebraic geometry, can not be applied in this case, and thus some other approach is required. Our objective is to apply the machinery of *generalized tensor decompositions*, and show universality and existence of depth efficiency in such RNNs.

## 2 Related work

Tensor methods have a rich history of successful application in machine learning. (Vasilescu & Terzopoulos, 2002), in their framework of TensorFaces, proposed to treat facial image data as multidimensional arrays and analyze them with tensor decompositions, which led to significant boost in face recognition accuracy. (Bailey & Aeron, 2017) employed higher-order co-occurence data and tensor factorization techniques to improve on word embeddings models. Tensor methods also allow to produce more accurate and robust recommender systems by taking into account a multifaceted nature of real environments (Frolov & Oseledets, 2017).

In recent years a great deal of work was done in applications of tensor calculus to both theoretical and practical aspects of deep learning algorithms. (Lebedev et al., 2015) represented filters in a convolutional network with CP decomposition (Harshman, 1970; Carroll & Chang, 1970) which allowed for much faster inference at the cost of a negligible drop in performance. (Novikov et al., 2015) proposed to use Tensor Train (TT) decomposition (Oseledets, 2011) to compress fully–connected layers of large neural networks while preserving their expressive power. Later on, TT was exploited to reduce the number of parameters and improve the performance of recurrent networks in long–term forecasting (Yu et al., 2017) and video classification (Yang et al., 2017) problems.

In addition to the practical benefits, tensor decompositions were used to analyze theoretical aspects of deep neural nets. (Cohen et al., 2016) investigated a connection between various network architectures and tensor decompositions, which made possible to compare their expressive power. Specifically, it was shown that CP and Hierarchial Tucker (Grasedyck, 2010) decompositions correspond to shallow networks and convolutional networks respectively. Recently, this analysis was extended by (Khrulkov et al., 2018) who showed that TT decomposition can be represented as a recurrent network with multiplicative connections. This specific form of RNNs was also empirically proved to provide a substantial performance boost over standard RNN models (Wu et al., 2016).

First results on the connection between tensor decompositions and neural networks were obtained for rather simple architectures, however, later on, they were extended in order to analyze more practical deep neural nets. It was shown that theoretical results can be generalized to a large class of CNNs with ReLU nonlinearities (Cohen & Shashua, 2016) and dilated convolutions (Cohen et al., 2018), providing valuable insights on how they can be improved. However, there is a missing piece in the whole picture as theoretical properties of more complex nonlinear RNNs have yet to be analyzed. In this paper, we elaborate on this problem and present new tools for conducting a theoretical analysis of such RNNs, specifically when rectifier nonlinearities are used.

## 3 ARCHITECTURES INSPIRED BY TENSOR DECOMPOSITIONS

Let us now recall the known results about the connection of tensor decompositions and multiplicative architectures, and then show how they are generalized in order to include networks with ReLU nonlinearities.

### 3.1 SCORE FUNCTIONS AND FEATURE TENSOR

Suppose that we are given a dataset of objects with a sequential structure, i.e. every object in the dataset can be written as

$$X = \left( \mathbf{x}^{(1)}, \mathbf{x}^{(2)}, \ldots, \mathbf{x}^{(T)} \right), \quad \mathbf{x}^{(t)} \in \mathbb{R}^N. \tag{1}$$

We also introduce a parametric *feature map* $f_\theta : \mathbb{R}^N \to \mathbb{R}^M$ which essentially preprocesses the data before it is fed into the network. Assumption 1 holds for many types of data, e.g. in the case of natural images we can cut them into rectangular patches which are then arranged into vectors $\mathbf{x}^{(t)}$. A typical choice for the feature map $f_\theta$ in this particular case is an affine map followed by a nonlinear activation: $f_\theta(\mathbf{x}) = \sigma(\mathbf{A}\mathbf{x} + \mathbf{b})$. To draw the connection between tensor decompositions and feature tensors we consider the following *score functions* (logits[1]):

$$\ell(X) = \langle \mathbf{\mathcal{W}}, \mathbf{\Phi}(X) \rangle = (\text{vec} \, \mathbf{\mathcal{W}})^\top \text{vec} \, \mathbf{\Phi}(X), \tag{2}$$

where $\mathbf{\mathcal{W}} \in \mathbb{R}^{M \times M \times \cdots \times M}$ is a trainable $T$–way weight tensor and $\mathbf{\Phi}(X) \in \mathbb{R}^{M \times M \times \cdots \times M}$ is a rank 1 *feature tensor*, defined as

$$\mathbf{\Phi}(X) = f_\theta(\mathbf{x}^{(1)}) \otimes f_\theta(\mathbf{x}^{(2)}) \cdots \otimes f_\theta(\mathbf{x}^{(T)}), \tag{3}$$

where we have used the operation of outer product $\otimes$, which is important in tensor calculus. For a tensor $\mathbf{\mathcal{A}}$ of order $N$ and a tensor $\mathbf{\mathcal{B}}$ of order $M$ their outer product $\mathbf{\mathcal{C}} = \mathbf{\mathcal{A}} \otimes \mathbf{\mathcal{B}}$ is a tensor of order $N + M$ defined as:

$$\mathbf{\mathcal{C}}_{i_1 i_2 \ldots i_N j_1 j_2 \ldots j_M} = \mathbf{\mathcal{A}}_{i_1 i_2 \cdots i_N} \mathbf{\mathcal{B}}_{j_1 j_2 \cdots j_M}. \tag{4}$$

---

[1]By *logits* we mean immediate outputs of the last hidden layer before applying nonlinearity. This term is adopted from classification tasks where neural network usually outputs *logits* and following softmax nonlinearity transforms them into valid probabilities.

It is known that equation 2 possesses the *universal approximation property* (it can approximate any function with any prescribed precision given sufficiently large $M$) under mild assumptions on $f_\theta$ (Cohen et al., 2016; Girosi & Poggio, 1990).

### 3.2 TENSOR DECOMPOSITIONS

Working the entire weight tensor $\mathcal{W}$ in eq. (2) is impractical for large $M$ and $T$, since it requires exponential in $T$ number of parameters. Thus, we compactly represent it using *tensor decompositions*, which will further lead to different neural network architectures, referred to as *tensor networks* (Cichocki et al., 2017).

**CP-decomposition** The most basic decomposition is the so-called Canonical (CP) decomposition (Harshman, 1970; Carroll & Chang, 1970) which is defined as follows

$$\mathcal{W} = \sum_{r=1}^{R} \lambda_r \mathbf{v}_r^{(1)} \otimes \mathbf{v}_r^{(2)} \cdots \otimes \mathbf{v}_r^{(T)}, \tag{5}$$

where $\mathbf{v}_r^{(t)} \in \mathbb{R}^M$ and minimal value of $R$ such that decomposition equation 5 exists is called *canonical rank of a tensor (CP–rank)*. By substituting eq. (5) into eq. (2) we find that

$$\ell(X) = \sum_{r=1}^{R} \lambda_r \left[ \langle f_\theta(\mathbf{x}^{(1)}), \mathbf{v}_r^{(1)} \rangle \otimes \cdots \otimes \langle f_\theta(\mathbf{x}^{(T)}), \mathbf{v}_r^{(T)} \rangle \right] = \sum_{r=1}^{R} \lambda_r \prod_{t=1}^{T} \langle f_\theta(\mathbf{x}^{(t)}), \mathbf{v}_r^{(t)} \rangle. \tag{6}$$

In the equation above, outer products $\otimes$ are taken between scalars and coincide with the ordinary products between two numbers. However, we would like to keep this notation as it will come in handy later, when we generalize tensor decompositions to include various nonlinearities.

**TT-decomposition** Another tensor decomposition is Tensor Train (TT) decomposition (Oseledets, 2011) which is defined as follows

$$\mathcal{W} = \sum_{r_1=1}^{R_1} \cdots \sum_{r_{T-1}=1}^{R_{T-1}} \mathbf{g}_{r_0 r_1}^{(1)} \otimes \mathbf{g}_{r_1 r_2}^{(2)} \otimes \cdots \otimes \mathbf{g}_{r_{T-1} r_T}^{(T)}, \tag{7}$$

where $\mathbf{g}_{r_{t-1} r_t}^{(t)} \in \mathbb{R}^M$ and $r_0 = r_T = 1$ by definition. If we gather vectors $\mathbf{g}_{r_{t-1} r_t}^{(t)}$ for all corresponding indices $r_{t-1} \in \{1, \ldots, R_{t-1}\}$ and $r_t \in \{1, \ldots, R_t\}$ we will obtain three–dimensional tensors $\mathcal{G}^{(t)} \in \mathbb{R}^{M \times R_{t-1} \times R_t}$ (for $t = 1$ and $t = T$ we will get matrices $\mathcal{G}^{(1)} \in \mathbb{R}^{M \times 1 \times R_1}$ and $\mathcal{G}^{(T)} \in \mathbb{R}^{M \times R_{T-1} \times 1}$). The set of all such tensors $\{\mathcal{G}^{(t)}\}_{t=1}^{T}$ is called *TT–cores* and minimal values of $\{R_t\}_{t=1}^{T-1}$ such that decomposition equation 7 exists are called *TT–ranks*. In the case of TT decomposition, the score function has the following form:

$$\ell(X) = \sum_{r_1=1}^{R_1} \cdots \sum_{r_{T-1}=1}^{R_{T-1}} \prod_{t=1}^{T} \langle f_\theta(\mathbf{x}^{(t)}), \mathbf{g}_{r_{t-1} r_t}^{(t)} \rangle. \tag{8}$$

### 3.3 CONNECTION BETWEEN TT AND RNN

Now we want to show that the score function for Tensor Train decomposition exhibits particular recurrent structure similar to that of RNN. We define the following *hidden states*:

$$
\begin{aligned}
\mathbf{h}^{(1)} \in \mathbb{R}^{R_1} : \mathbf{h}_{r_1}^{(1)} &= \langle f_\theta(\mathbf{x}^{(1)}), \mathbf{g}_{r_0 r_1}^{(1)} \rangle, \\
\mathbf{h}^{(t)} \in \mathbb{R}^{R_t} : \mathbf{h}_{r_t}^{(t)} &= \sum_{r_{t-1}=1}^{R_{t-1}} \langle f_\theta(\mathbf{x}^{(t)}), \mathbf{g}_{r_{t-1} r_t}^{(t)} \rangle \mathbf{h}_{r_{t-1}}^{(t-1)} \quad t = 2, \ldots, T.
\end{aligned} \tag{9}
$$

Such definition of hidden states allows for more compact form of the score function.

**Lemma 3.1.** *Under the notation introduced in eq.* (9)*, the score function can be written as*

$$\ell(X) = \mathbf{h}^{(T)} \in \mathbb{R}^1.$$

Proof of Lemma 3.1 as well as the proofs of our main results from Section 5 were moved to Appendix A due to limited space.

Note that with a help of TT–cores we can rewrite eq. (9) in a more convenient index form:

$$\mathbf{h}_k^{(t)} = \sum_{i,j} \boldsymbol{\mathcal{G}}_{ijk}^{(t)} \, f_\theta(\mathbf{x}^{(t)})_i \, \mathbf{h}_j^{(t-1)} = \sum_{i,j} \boldsymbol{\mathcal{G}}_{ijk}^{(t)} \left[ f_\theta(\mathbf{x}^{(t)}) \otimes \mathbf{h}^{(t-1)} \right]_{ij}, \quad k = 1, \dots, R_t, \quad (10)$$

where the operation of tensor contraction is used. Combining all weights from $\boldsymbol{\mathcal{G}}^{(t)}$ and $f_\theta(\cdot)$ into a single variable $\Theta_{\boldsymbol{\mathcal{G}}}^{(t)}$ and denoting the composition of feature map, outer product, and contraction as $g : \mathbb{R}^{R_{t-1}} \times \mathbb{R}^N \times \mathbb{R}^{N \times R_{t-1} \times R_t} \to \mathbb{R}^{R_t}$ we arrive at the following vector form:

$$\mathbf{h}^{(t)} = g(\mathbf{h}^{(t-1)}, \mathbf{x}^{(t)}; \Theta_{\boldsymbol{\mathcal{G}}}^{(t)}), \quad \mathbf{h}^{(t)} \in \mathbb{R}^{R_t}. \quad (11)$$

This equation can be considered as a generalization of hidden state equation for Recurrent Neural Networks as here all hidden states $\mathbf{h}^{(t)}$ may in general have different dimensionalities and weight tensors $\Theta_{\boldsymbol{\mathcal{G}}}^{(t)}$ depend on the time step. However, if we set $R = R_1 = \dots = R_{T-1}$ and $\boldsymbol{\mathcal{G}} = \boldsymbol{\mathcal{G}}^{(2)} = \dots = \boldsymbol{\mathcal{G}}^{(T-1)}$ we will get simplified hidden state equation used in standard recurrent architectures:

$$\mathbf{h}^{(t)} = g(\mathbf{h}^{(t-1)}, \mathbf{x}^{(t)}; \Theta_{\boldsymbol{\mathcal{G}}}), \quad \mathbf{h}^{(t)} \in \mathbb{R}^R, \quad t = 2, \dots, T-1. \quad (12)$$

Note that this equation is applicable to all hidden states except for the first $\mathbf{h}^{(1)} = \boldsymbol{\mathcal{G}}^{(1)} f_\theta(\mathbf{x}^{(1)})$ and for the last $\mathbf{h}^{(T)} = f_\theta^\top(\mathbf{x}^{(T)}) \boldsymbol{\mathcal{G}}^{(T)} \mathbf{h}^{(T-1)}$, due to two–dimensional nature of the corresponding TT–cores. However, we can always pad the input sequence with two auxiliary vectors $\mathbf{x}^{(0)}$ and $\mathbf{x}^{(T+1)}$ to get full compliance with the standard RNN structure. Figure 1 depicts tensor network induced by TT decomposition with cores $\{\boldsymbol{\mathcal{G}}^{(t)}\}_{t=1}^T$.

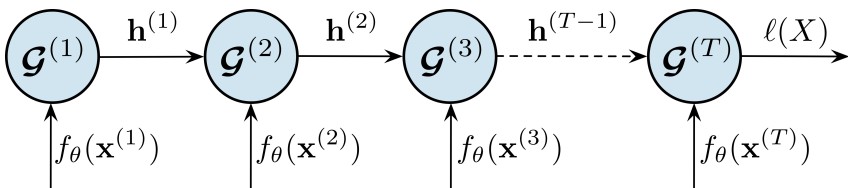

Figure 1: Neural network architecture which corresponds to recurrent TT–Network.

# 4 GENERALIZED TENSOR NETWORKS

## 4.1 GENERALIZED OUTER PRODUCT

In the previous section we showed that tensor decompositions correspond to neural networks of specific structure, which are simplified versions of those used in practice as they contain multiplicative nonlinearities only. One possible way to introduce more practical nonlinearities is to replace outer product $\otimes$ in eq. (6) and eq. (10) with a generalized operator $\otimes_\xi$ in analogy to kernel methods when scalar product is replaced by nonlinear kernel function. Let $\xi : \mathbb{R} \times \mathbb{R} \to \mathbb{R}$ be an associative and commutative binary operator ($\forall x, y, z \in \mathbb{R} : \xi(\xi(x, y), z) = \xi(x, \xi(y, z))$ and $\forall x, y \in \mathbb{R} : \xi(x, y) = \xi(y, x)$). Note that this operator easily generalizes to the arbitrary number of operands due to associativity. For a tensor $\boldsymbol{\mathcal{A}}$ of order $N$ and a tensor $\boldsymbol{\mathcal{B}}$ of order $M$ we define their generalized outer product $\boldsymbol{\mathcal{C}} = \boldsymbol{\mathcal{A}} \otimes_\xi \boldsymbol{\mathcal{B}}$ as an $(N + M)$ order tensor with entries given by:

$$\boldsymbol{\mathcal{C}}_{i_1 \dots i_N j_1 \dots j_M} = \xi\left(\boldsymbol{\mathcal{A}}_{i_1 \dots i_N}, \boldsymbol{\mathcal{B}}_{j_1 \dots j_M}\right). \quad (13)$$

Now we can replace $\otimes$ in eqs. (6) and (10) with $\otimes_\xi$ and get networks with various nonlinearities. For example, if we take $\xi(x, y) = \max(x, y, 0)$ we will get an RNN with rectifier nonlinearities; if we take $\xi(x, y) = \ln(e^x + e^y)$ we will get an RNN with softplus nonlinearities; if we take $\xi(x, y) = xy$ we will get a simple RNN defined in the previous section. Concretely, we will analyze the following networks.

**Generalized shallow network with $\xi$–nonlinearity**

- Score function:

$$
\begin{aligned}
\ell(X) &= \sum_{r=1}^{R} \lambda_r \left[ \langle f_\theta(\mathbf{x}^{(1)}), \mathbf{v}_r^{(1)} \rangle \otimes_\xi \cdots \otimes_\xi \langle f_\theta(\mathbf{x}^{(T)}), \mathbf{v}_r^{(T)} \rangle \right] \\
&= \sum_{r=1}^{R} \lambda_r \xi \left( \langle f_\theta(\mathbf{x}^{(1)}), \mathbf{v}_r^{(1)} \rangle, \ldots, \langle f_\theta(\mathbf{x}^{(T)}), \mathbf{v}_r^{(T)} \rangle \right)
\end{aligned}
\tag{14}
$$

- Parameters of the network:

$$
\Theta = \left( \{\lambda_r\}_{r=1}^{R} \in \mathbb{R}, \{\mathbf{v}_r^{(t)}\}_{r=1,t=1}^{R,T} \in \mathbb{R}^M \right)
\tag{15}
$$

**Generalized RNN with $\xi$–nonlinearity**

- Score function:

$$
\mathbf{h}_k^{(t)} = \sum_{i,j} \boldsymbol{\mathcal{G}}_{ijk}^{(t)} \left[ \mathbf{C}^{(t)} f_\theta(\mathbf{x}^{(t)}) \otimes_\xi \mathbf{h}^{(t-1)} \right]_{ij} = \sum_{i,j} \boldsymbol{\mathcal{G}}_{ijk}^{(t)} \, \xi \left( [\mathbf{C}^{(t)} f_\theta(\mathbf{x}^{(t)})]_i, \mathbf{h}_j^{(t-1)} \right)
$$

$$
\ell(X) = \mathbf{h}^{(T)}
\tag{16}
$$

- Parameters of the network:

$$
\Theta = \left( \{\mathbf{C}^{(t)}\}_{t=1}^{T} \in \mathbb{R}^{L \times M}, \{\boldsymbol{\mathcal{G}}^{(t)}\}_{t=1}^{T} \in \mathbb{R}^{L \times R_{t-1} \times R_t} \right)
\tag{17}
$$

Note that in eq. (16) we have introduced the matrices $\mathbf{C}^{(t)}$ acting on the input states. The purpose of this modification is to obtain the plausible property of generalized shallow networks being able to be represented as generalized RNNs of width 1 (i.e., with all $R_i = 1$) for an arbitrary nonlinearity $\xi$. In the case of $\xi(x, y) = xy$, the matrices $\mathbf{C}^{(t)}$ were not necessary, since they can be simply absorbed by $\boldsymbol{\mathcal{G}}^{(t)}$ via tensor contraction (see Appendix A for further clarification on these points).

**Initial hidden state**    Note that generalized RNNs require some choice of the initial hidden state $\mathbf{h}^{(0)}$. We find that it is convenient both for theoretical analysis and in practice to initialize $\mathbf{h}^{(0)}$ as *unit* of the operator $\xi$, i.e. such an element $u$ that $\xi(x, y, u) = \xi(x, y) \; \forall x, y \in \mathbb{R}$. Henceforth, we will assume that such an element exists (e.g., for $\xi(x, y) = \max(x, y, 0)$ we take $u = 0$, for $\xi(x, y) = xy$ we take $u = 1$), and set $\mathbf{h}^{(0)} = u$. For example, in eq. (9) it was implicitly assumed that $\mathbf{h}^{(0)} = 1$.

## 4.2    Grid tensors

Introduction of generalized outer product allows us to investigate RNNs with wide class of nonlinear activation functions, especially ReLU. While this change looks appealing from the practical viewpoint, it complicates following theoretical analysis, as the transition from obtained networks back to tensors is not straightforward.

In the discussion above, every tensor network had corresponding weight tensor $\boldsymbol{\mathcal{W}}$ and we could compare expressivity of associated score functions by comparing some properties of this tensors, such as ranks (Khrulkov et al., 2018; Cohen et al., 2016). This method enabled comprehensive analysis of score functions, as it allows us to calculate and compare their values for all possible input sequences $X = \left( \mathbf{x}^{(1)}, \ldots, \mathbf{x}^{(T)} \right)$. Unfortunately, we can not apply it in case of generalized tensor networks, as the replacement of standard outer product $\otimes$ with its generalized version $\otimes_\xi$ leads to the loss of conformity between tensor networks and weight tensors. Specifically, not for every generalized tensor network with corresponding score function $\ell(X)$ now exists a weight tensor $\boldsymbol{\mathcal{W}}$ such that $\ell(X) = \langle \boldsymbol{\mathcal{W}}, \boldsymbol{\Phi}(X) \rangle$. Also, such properties as *universality* no longer hold automatically and we have to prove them separately. Indeed as it was noticed in (Cohen & Shashua, 2016) shallow networks with $\xi(x, y) = \max(x, 0) + \max(y, 0)$ no longer have the universal approximation property. In order to conduct proper theoretical analysis, we adopt the apparatus of so-called *grid tensors*, first introduced in (Cohen & Shashua, 2016).

Given a set of fixed vectors $\mathbb{X} = \left\{ \mathbf{x}^{(1)}, \ldots, \mathbf{x}^{(M)} \right\}$ referred to as *templates*, the grid tensor of $\mathbb{X}$ is defined to be the tensor of order $T$ and dimension $M$ in each mode, with entries given by:

$$\mathbf{\Gamma}^{\ell}(\mathbb{X})_{i_1 i_2 \ldots i_T} = \ell(X), \quad X = \left( \mathbf{x}^{(i_1)}, \mathbf{x}^{(i_2)}, \ldots, \mathbf{x}^{(i_T)} \right), \tag{18}$$

where each index $i_t$ can take values from $\{1, \ldots, M\}$, i.e. we evaluate the score function on every possible input assembled from the template vectors $\{\mathbf{x}^{(i)}\}_{i=1}^{M}$. To put it simply, we previously considered the equality of score functions represented by tensor decomposition and tensor network on set of all possible input sequences $X = \left( \mathbf{x}^{(1)}, \ldots, \mathbf{x}^{(T)} \right)$, $\mathbf{x}^{(t)} \in \mathbb{R}^N$, and now we restricted this set to exponentially large but finite grid of sequences consisting of template vectors only.

Define the matrix $\mathbf{F} \in \mathbb{R}^{M \times M}$ which holds the values taken by the representation function $f_\theta : \mathbb{R}^N \to \mathbb{R}^M$ on the selected templates $\mathbb{X}$:

$$\mathbf{F} \triangleq \begin{bmatrix} f_\theta(\mathbf{x}^{(1)}) & f_\theta(\mathbf{x}^{(2)}) & \ldots & f_\theta(\mathbf{x}^{(M)}) \end{bmatrix}^{\top}. \tag{19}$$

Using the matrix $\mathbf{F}$ we note that the grid tensor of generalized shallow network has the following form (see Appendix A for derivation):

$$\mathbf{\Gamma}^{\ell}(\mathbb{X}) = \sum_{r=1}^{R} \lambda_r \left( \mathbf{F} \mathbf{v}_r^{(1)} \right) \otimes_{\xi} \left( \mathbf{F} \mathbf{v}_r^{(2)} \right) \otimes_{\xi} \cdots \otimes_{\xi} \left( \mathbf{F} \mathbf{v}_r^{(T)} \right). \tag{20}$$

Construction of the grid tensor for generalized RNN is a bit more involved. We find that its grid tensor $\mathbf{\Gamma}^{\ell}(\mathbb{X})$ can be computed recursively, similar to the hidden state in the case of a single input sequence. The exact formulas turned out to be rather cumbersome and we moved them to Appendix A.

## 5 MAIN RESULTS

With grid tensors at hand we are ready to compare the expressive power of generalized RNNs and generalized shallow networks. In the further analysis, we will assume that $\xi(x, y) = \max(x, y, 0)$, i.e., we analyze RNNs and shallow networks with *rectifier nonlinearity*. However, we need to make two additional assumptions. First of all, similarly to (Cohen & Shashua, 2016) we fix some templates $\mathbb{X}$ such that values of the score function outside of the grid generated by $\mathbb{X}$ are *irrelevant for classification* and call them *covering* templates. It was argued that for image data values of $M$ of order 100 are sufficient (corresponding covering template vectors may represent Gabor filters). Secondly, we assume that the feature matrix $\mathbf{F}$ is invertible, which is a reasonable assumption and in the case of $f_\theta(\mathbf{x}) = \sigma(\mathbf{A}\mathbf{x} + \mathbf{b})$ for any distinct template vectors $\mathbb{X}$ the parameters $\mathbf{A}$ and $\mathbf{b}$ can be chosen in such a way that the matrix $\mathbf{F}$ is invertible.

### 5.1 UNIVERSALITY

As was discussed in section 4.2 we can no longer use standard algebraic techniques to verify universality of tensor based networks. Thus, our first result states that generalized RNNs with $\xi(x, y) = \max(x, y, 0)$ are *universal* in a sense that any tensor of order $T$ and size of each mode being $m$ can be realized as a grid tensor of such RNN (and similarly of a generalized shallow network).

**Theorem 5.1** (Universality). *Let $\mathcal{H} \in \mathbb{R}^{M \times M \times \cdots \times M}$ be an arbitrary tensor of order $T$. Then there exist a **generalized shallow network** and a **generalized RNN** with rectifier nonlinearity $\xi(x, y) = \max(x, y, 0)$ such that grid tensor of each of the networks coincides with $\mathcal{H}$.*

Part of Theorem 5.1 which corresponds to generalized shallow networks readily follows from (Cohen & Shashua, 2016, Claim 4). In order to prove the statement for the RNNs the following two lemmas are used.

**Lemma 5.1.** *Given two generalized RNNs with grid tensors $\mathbf{\Gamma}^{\ell_A}(\mathbb{X})$, $\mathbf{\Gamma}^{\ell_B}(\mathbb{X})$, and arbitrary $\xi$-nonlinearity, there exists a generalized RNN with grid tensor $\mathbf{\Gamma}^{\ell_C}(\mathbb{X})$ satisfying*

$$\mathbf{\Gamma}^{\ell_C}(\mathbb{X}) = a\mathbf{\Gamma}^{\ell_A}(\mathbb{X}) + b\mathbf{\Gamma}^{\ell_B}(\mathbb{X}), \quad \forall a, b \in \mathbb{R}.$$

This lemma essentially states that the collection of grid tensors of generalized RNNs with any nonlinearity is closed under taking arbitrary linear combinations. Note that the same result clearly holds for generalized shallow networks because they are linear combinations of rank 1 shallow networks by definition.

**Lemma 5.2.** *Let* $\boldsymbol{\mathcal{E}}^{(j_1 j_2 \ldots j_T)}$ *be an arbitrary one–hot tensor, defined as*

$$\boldsymbol{\mathcal{E}}^{(j_1 j_2 \ldots j_T)}_{i_1 i_2 \ldots i_T} = \begin{cases} 1, & j_t = i_t \quad \forall t \in \{1, \ldots, T\}, \\ 0, & otherwise. \end{cases}$$

*Then there exists a generalized RNN with rectifier nonlinearities such that its grid tensor satisfies*

$$\boldsymbol{\Gamma}^\ell(\mathbb{X}) = \boldsymbol{\mathcal{E}}^{(j_1 j_2 \ldots j_T)}.$$

This lemma states that in the special case of rectifier nonlinearity $\xi(x, y) = \max(x, y, 0)$ any *basis* tensor can be realized by some generalized RNN.

**Proof of Theorem 5.1.** By Lemma 5.2 for each one–hot tensor $\boldsymbol{\mathcal{E}}^{(i_1 i_2 \ldots i_T)}$ there exists a generalized RNN with rectifier nonlinearities, such that its grid tensor coincides with this tensor. Thus, by Lemma 5.1 we can construct an RNN with

$$\boldsymbol{\Gamma}^\ell(\mathbb{X}) = \sum_{i_1, i_2, \ldots, i_T} \boldsymbol{\mathcal{H}}_{i_1 i_2 \ldots i_d} \boldsymbol{\mathcal{E}}^{(i_1 i_2 \ldots i_T)} = \boldsymbol{\mathcal{H}}.$$

For generalized shallow networks with rectifier nonlinearities see the proof of (Cohen & Shashua, 2016, Claim 4). □

The same result regarding networks with product nonlinearities considered in (Khrulkov et al., 2018) directly follows from the well–known properties of tensor decompositions (see Appendix A).

We see that at least with such nonlinearities as $\xi(x, y) = \max(x, y, 0)$ and $\xi(x, y) = xy$ all the networks under consideration are universal and can represent any possible grid tensor. Now let us head to a discussion of *expressivity* of these networks.

## 5.2 EXPRESSIVITY

As was discussed in the introduction, expressivity refers to the ability of some class of networks to represent the same functions as some other class much more compactly. In our case the parameters defining *size* of networks are *ranks* of the decomposition, i.e. in the case of generalized RNNs ranks determine the size of the hidden state, and in the case of generalized shallow networks rank determines the width of a network. It was proven in (Cohen et al., 2016; Khrulkov et al., 2018) that ConvNets and RNNs with multiplicative nonlinearities are *exponentially* more expressive than the equivalent shallow networks: shallow networks of exponentially large width are required to realize the same score functions as computed by these deep architectures. Similarly to the case of ConvNets (Cohen & Shashua, 2016), we find that expressivity of generalized RNNs with rectifier nonlinearity holds only partially, as discussed in the following two theorems. For simplicity, we assume that $T$ is even.

**Theorem 5.2** (Expressivity 1). *For every value of $R$ there exists a generalized RNN with ranks $\leq R$ and rectifier nonlinearity which is exponentially more efficient than shallow networks, i.e., the corresponding grid tensor may be realized only by a shallow network with rectifier nonlinearity of width at least $\frac{2}{MT} \min(M, R)^{T/2}$.*

This result states that at least for some subset of generalized RNNs expressivity holds: exponentially wide shallow networks are required to realize the same grid tensor. Proof of the theorem is rather straightforward: we explicitly construct an example of such RNN which satisfies the following description. Given an arbitrary input sequence $X = \left(\mathbf{x}^{(1)}, \ldots \mathbf{x}^{(T)}\right)$ assembled from the templates, these networks (if $M = R$) produce 0 if $X$ has the property that $\mathbf{x}^{(1)} = \mathbf{x}^{(2)}, \mathbf{x}^{(3)} = \mathbf{x}^{(4)}, \ldots, \mathbf{x}^{(T-1)} = \mathbf{x}^{(T)}$, and 1 in every other case, i.e. they measure *pairwise similarity* of the input vectors. A precise proof is given in Appendix A.

In the case of multiplicative RNNs (Khrulkov et al., 2018) *almost every* network possessed this property. This is not the case, however, for generalized RNNs with rectifier nonlinearities.

**Theorem 5.3** (Expressivity 2). *For every value of $R$ there exists an open set (which thus has positive measure) of generalized RNNs with rectifier nonlinearity $\xi(x,y) = \max(x,y,0)$, such that for each RNN in this open set the corresponding grid tensor can be realized by a rank $1$ shallow network with rectifier nonlinearity.*

In other words, for every rank $R$ we can find a set of generalized RNNs of positive measure such that the property of expressivity does not hold. In the numerical experiments in Section 6 and Appendix A we validate whether this can be observed in practice, and find that the probability of obtaining CP–ranks of polynomial size becomes negligible with large $T$ and $R$. Proof of Theorem 5.3 is provided in Appendix A.

**Shared case**    Note that all the RNNs used in practice have *shared weights*, which allows them to process sequences of arbitrary length. So far in the analysis we have not made such assumptions about RNNs (i.e., $\mathcal{G}^{(2)} = \cdots = \mathcal{G}^{(T-1)}$). By imposing this constraint, we lose the property of universality; however, we believe that the statements of Theorems 5.2 and 5.3 still hold (without requiring that shallow networks also have shared weights). Note that the example constructed in the proof of Theorem 5.3 already has this property, and for Theorem 5.2 we provide numerical evidence in Appendix A.

## 6    EXPERIMENTS

In this section, we study if our theoretical findings are supported by experimental data. In particular, we investigate whether generalized tensor networks can be used in practical settings, especially in problems typically solved by RNNs (such as natural language processing problems). Secondly, according to Theorem 5.3 for some subset of RNNs the equivalent shallow network may have a low rank. To get a grasp of how strong this effect might be in practice we numerically compute an estimate for this rank in various settings.

**Performance**    For the first experiment, we use two computer vision datasets MNIST (LeCun et al., 1990) and CIFAR–10 (Krizhevsky & Hinton, 2009), and natural language processing dataset for sentiment analysis IMDB (Maas et al., 2011). For the first two datasets, we cut natural images into rectangular patches which are then arranged into vectors $\mathbf{x}^{(t)}$ (similar to (Khrulkov et al., 2018)) and for IMDB dataset the input data already has the desired sequential structure.

Figure 2 depicts test accuracy on IMDB dataset for generalized shallow networks and RNNs with rectifier nonlinearity. We see that generalized shallow network of much higher rank is required to get the level of performance close to that achievable by generalized RNN. Due to limited space, we have moved the results of the experiments on the visual datasets to Appendix B.

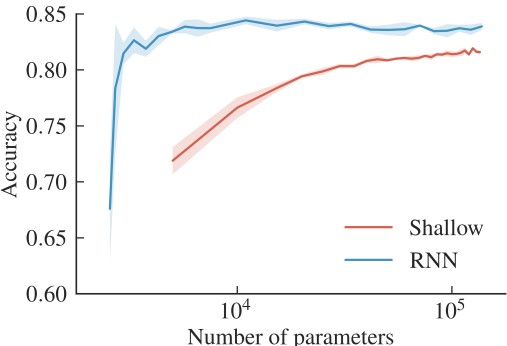
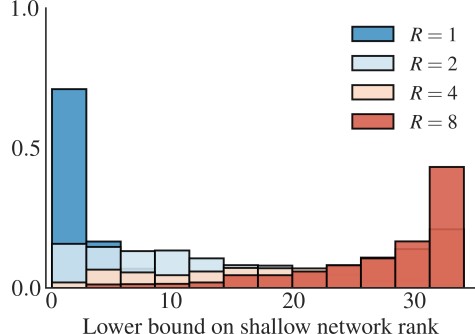

Figure 2: Test accuracy on IMDB dataset for generalized RNNs and generalized shallow networks with respect to the total number of parameters ($M = 50$, $T = 100$, $\xi(x,y) = \max(x,y,0)$).

Figure 3: Distribution of lower bounds on the rank of generalized shallow networks equivalent to randomly generated generalized RNNs of ranks $1, 2, 4, 8$ ($M = 10$, $T = 6$).

**Expressivity**   For the second experiment we generate a number of generalized RNNs with different values of TT-rank $r$ and calculate a lower bound on the rank of shallow network necessary to realize the same grid tensor (to estimate the rank we use the same technique as in the proof of Theorem 5.2). Figure 3 shows that for different values of $R$ and generalized RNNs of the corresponding rank there exist shallow networks of rank 1 realizing the same grid tensor, which agrees well with Theorem 5.3. This result looks discouraging, however, there is also a positive observation. While increasing rank of generalized RNNs, more and more corresponding shallow networks will necessarily have exponentially higher rank. In practice we usually deal with RNNs of $R = 10^2 - 10^3$ (dimension of hidden states), thus we may expect that effectively any function besides negligible set realized by generalized RNNs can be implemented only by exponentially wider shallow networks. The numerical results for the case of shared cores and other nonlinearities are given in Appendix B.

## 7   Conclusion

In this paper, we sought a more complete picture of the connection between Recurrent Neural Networks and Tensor Train decomposition, one that involves various nonlinearities applied to hidden states. We showed how these nonlinearities could be incorporated into network architectures and provided complete theoretical analysis on the particular case of rectifier nonlinearity, elaborating on points of generality and expressive power. We believe our results will be useful to advance theoretical understanding of RNNs. In future work, we would like to extend the theoretical analysis to most competitive in practice architectures for processing sequential data such as LSTMs and attention mechanisms.

## Acknowledgements

We would like to thank Andrzej Cichocki for constructive discussions during the preparation of the manuscript and anonymous reviewers for their valuable feedback. This work was supported by the Ministry of Education and Science of the Russian Federation (grant 14.756.31.0001).

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

## A   PROOFS

**Lemma 3.1.** *Under the notation introduced in eq. (9), the score function can be written as*

$$\ell(X) = \mathbf{h}^{(T)} \in \mathbb{R}^1.$$

**Proof.**

$$l(X) = \sum_{r_1=1}^{R_1} \cdots \sum_{r_{T-1}=1}^{R_{T-1}} \prod_{t=1}^{T} \langle f_\theta(\mathbf{x}^{(t)}), \mathbf{g}_{r_{t-1}r_t}^{(t)} \rangle$$

$$= \sum_{r_1=1}^{R_1} \cdots \sum_{r_{T-1}=1}^{R_{T-1}} \prod_{t=2}^{T} \langle f_\theta(\mathbf{x}^{(t)}), \mathbf{g}_{r_{t-1}r_t}^{(t)} \rangle \underbrace{\langle f_\theta(\mathbf{x}^{(1)}), \mathbf{g}_{r_0 r_1}^{(1)} \rangle}_{\mathbf{h}_{r_1}^{(1)}}$$

$$= \sum_{r_{T-1}=1}^{R_{T-1}} \cdots \sum_{r_1=1}^{R_1} \prod_{t=2}^{T} \langle f_\theta(\mathbf{x}^{(t)}), \mathbf{g}_{r_{t-1}r_t}^{(t)} \rangle \mathbf{h}_{r_1}^{(1)}$$

$$= \sum_{r_{T-1}=1}^{R_{T-1}} \cdots \sum_{r_2=1}^{R_2} \prod_{t=3}^{T} \langle f_\theta(\mathbf{x}^{(t)}), \mathbf{g}_{r_{t-1}r_t}^{(t)} \rangle \underbrace{\sum_{r_1=1}^{r_1} \langle f_\theta(\mathbf{x}^{(2)}), \mathbf{g}_{r_1 r_2}^{(2)} \rangle \mathbf{h}_{r_1}^{(1)}}_{\mathbf{h}_{r_2}^{(2)}}$$

$$= \sum_{r_{T-1}=1}^{R_{T-1}} \cdots \sum_{r_2=1}^{R_2} \prod_{t=3}^{T} \langle f_\theta(\mathbf{x}^{(t)}), \mathbf{g}_{r_{t-1}r_t}^{(t)} \rangle \mathbf{h}_{r_2}^{(2)}$$

$$= \dots$$

$$= \sum_{r_{T-1}=1}^{R_{T-1}} \langle f_\theta(\mathbf{x}^{(T)}), \mathbf{g}_{r_{T-1}r_T}^{(T)} \rangle \mathbf{h}_{r_{T-1}}^{(T-1)} = \mathbf{h}_{r_T}^{(T)} = \mathbf{h}^{(T)}.$$

$$\square$$

**Proposition A.1.** *If we replace the generalized outer product $\otimes_\xi$ in eq. (16) with the standard outer product $\otimes$, we can subsume matrices $\mathbf{C}^{(t)}$ into tensors $\boldsymbol{\mathcal{G}}^{(t)}$ without loss of generality.*

**Proof.** Let us rewrite hidden state equation eq. (16) after transition from $\otimes_\xi$ to $\otimes$:

$$\mathbf{h}_k^{(t)} = \sum_{i,j} \boldsymbol{\mathcal{G}}_{ijk}^{(t)} \left[ \mathbf{C}^{(t)} f_\theta(\mathbf{x}^{(t)}) \otimes \mathbf{h}^{(t-1)} \right]_{ij}$$

$$= \sum_{i,j} \boldsymbol{\mathcal{G}}_{ijk}^{(t)} \sum_l \mathbf{C}_{il}^{(t)} f_\theta(\mathbf{x}^{(t)})_l \mathbf{h}_j^{(t-1)} \qquad \left\{ \tilde{\boldsymbol{\mathcal{G}}}_{ljk}^{(t)} = \sum_i \boldsymbol{\mathcal{G}}_{ijk}^{(t)} \mathbf{C}_{il}^{(t)} \right\}$$

$$= \sum_{l,j} \tilde{\boldsymbol{\mathcal{G}}}_{ljk}^{(t)} f_\theta(\mathbf{x}^{(t)})_l \mathbf{h}_j^{(t-1)}$$

$$= \sum_{l,j} \tilde{\boldsymbol{\mathcal{G}}}_{ljk}^{(t)} \left[ f_\theta(\mathbf{x}^{(t)}) \otimes \mathbf{h}^{(t-1)} \right]_{lj}.$$

We see that the obtained expression resembles those presented in eq. (10) with TT-cores $\boldsymbol{\mathcal{G}}^{(t)}$ replaced by $\tilde{\boldsymbol{\mathcal{G}}}^{(t)}$ and thus all the reasoning applied in the absence of matrices $\mathbf{C}^{(t)}$ holds valid.       $\square$

**Proposition A.2.** *Grid tensor of generalized shallow network has the following form (eq. (20)):*

$$\boldsymbol{\Gamma}^\ell(\mathbb{X}) = \sum_{r=1}^{R} \lambda_r \left( \mathbf{F} \mathbf{v}_r^{(1)} \right) \otimes_\xi \left( \mathbf{F} \mathbf{v}_r^{(2)} \right) \otimes_\xi \cdots \otimes_\xi \left( \mathbf{F} \mathbf{v}_r^{(T)} \right).$$

**Proof.** Let $X = \left(\mathbf{x}^{(i_1)}, \mathbf{x}^{(i_2)}, \ldots, \mathbf{x}^{(i_T)}\right)$ denote an arbitrary sequence of *templates*. Corresponding element of the grid tensor defined in eq. (20) has the following form:

$$
\begin{aligned}
\mathbf{\Gamma}^\ell(\mathbb{X})_{i_1 i_2 \ldots i_T} &= \sum_{r=1}^R \lambda_r \left[ \left(\mathbf{F}\mathbf{v}_r^{(1)}\right) \otimes_\xi \left(\mathbf{F}\mathbf{v}_r^{(2)}\right) \otimes_\xi \cdots \otimes_\xi \left(\mathbf{F}\mathbf{v}_r^{(T)}\right) \right]_{i_1 i_2 \ldots i_T} \\
&= \sum_{r=1}^R \lambda_r \left(\mathbf{F}\mathbf{v}_r^{(1)}\right)_{i_1} \otimes_\xi \left(\mathbf{F}\mathbf{v}_r^{(2)}\right)_{i_2} \otimes_\xi \cdots \otimes_\xi \left(\mathbf{F}\mathbf{v}_r^{(T)}\right)_{i_T} \\
&= \sum_{r=1}^R \lambda_r \xi \left( \langle f_\theta(\mathbf{x}^{(i_1)}), \mathbf{v}_r^{(1)} \rangle, \ldots, \langle f_\theta(\mathbf{x}^{(i_T)}), \mathbf{v}_r^{(T)} \rangle \right) = \ell(X).
\end{aligned}
$$

$\square$

**Proposition A.3.** *Grid tensor of a generalized RNN has the following form:*

$$
\begin{aligned}
\mathbf{\Gamma}^{\ell,0}(\mathbb{X}) &= \mathbf{h}^{(0)} \in \mathbb{R}^1, \\
\mathbf{\Gamma}^{\ell,1}(\mathbb{X})_{km_1} &= \sum_{i,j} \boldsymbol{\mathcal{G}}_{ijk}^{(1)} \left( \mathbf{C}^{(1)} \mathbf{F}^\top \otimes_\xi \mathbf{\Gamma}^{\ell,0} \right)_{im_1 j} \in \mathbb{R}^{R_1 \times M}, \\
\mathbf{\Gamma}^{\ell,2}(\mathbb{X})_{km_1 m_2} &= \sum_{i,j} \boldsymbol{\mathcal{G}}_{ijk}^{(2)} \left( \mathbf{C}^{(2)} \mathbf{F}^\top \otimes_\xi \mathbf{\Gamma}^{\ell,1} \right)_{im_2 j m_1} \in \mathbb{R}^{R_2 \times M \times M}, \\
&\cdots \\
\mathbf{\Gamma}^{\ell,T}(\mathbb{X})_{km_1 m_2 \ldots m_T} &= \sum_{i,j} \boldsymbol{\mathcal{G}}_{ijk}^{(T)} \left( \mathbf{C}^{(T)} \mathbf{F}^\top \otimes_\xi \mathbf{\Gamma}^{\ell,T-1} \right)_{im_T j m_1 \ldots m_{T-1}} \in \mathbb{R}^{1 \times M \times M \times \cdots \times M}, \\
\mathbf{\Gamma}^\ell(\mathbb{X}) &= \mathbf{\Gamma}^{\ell,T}(\mathbb{X})_{1,:,:,\ldots,:}
\end{aligned}
$$

(21)

**Proof.** Proof is similar to that of Proposition A.2 and uses eq. (16) to compute the elements of the grid tensor. $\square$

**Lemma 5.1.** *Given two generalized RNNs with grid tensors $\mathbf{\Gamma}^{\ell_A}(\mathbb{X})$, $\mathbf{\Gamma}^{\ell_B}(\mathbb{X})$, and arbitrary $\xi$-nonlinearity, there exists a generalized RNN with grid tensor $\mathbf{\Gamma}^{\ell_C}(\mathbb{X})$ satisfying*

$$
\mathbf{\Gamma}^{\ell_C}(\mathbb{X}) = a\mathbf{\Gamma}^{\ell_A}(\mathbb{X}) + b\mathbf{\Gamma}^{\ell_B}(\mathbb{X}), \quad \forall a, b \in \mathbb{R}.
$$

**Proof.** Let these RNNs be defined by the weight parameters

$$
\Theta_A = \left( \{\mathbf{C}_A^{(t)}\}_{t=1}^T \in \mathbb{R}^{L_A \times M}, \{\boldsymbol{\mathcal{G}}_A^{(t)}\}_{t=1}^T \in \mathbb{R}^{L_A \times R_{t-1,A} \times R_{t,A}} \right),
$$

and

$$
\Theta_B = \left( \{\mathbf{C}_B^{(t)}\}_{t=1}^T \in \mathbb{R}^{L_B \times M}, \{\boldsymbol{\mathcal{G}}_B^{(t)}\}_{t=1}^T \in \mathbb{R}^{L_B \times R_{t-1,B} \times R_{t,B}} \right).
$$

We claim that the desired grid tensor is given by the RNN with the following weight settings.

$$\mathbf{C}_C^{(t)} \in \mathbb{R}^{(L_A+L_B)\times M}$$

$$\mathbf{C}_C^{(t)} = \begin{bmatrix} \mathbf{C}_A^{(t)} \\ \mathbf{C}_B^{(t)} \end{bmatrix}$$

$$\boldsymbol{\mathcal{G}}_C^{(1)} \in \mathbb{R}^{(L_A+L_B)\times 1 \times (R_{t,A}+R_{t,B})}$$

$$[\boldsymbol{\mathcal{G}}_C^{(1)}]_{i,:,:} = \begin{cases} \begin{bmatrix} [\boldsymbol{\mathcal{G}}_A^{(1)}]_{i,:,:} & 0 \end{bmatrix}, & i \in \{1,\dots,L_A\} \\[2ex] \begin{bmatrix} 0 & [\boldsymbol{\mathcal{G}}_B^{(1)}]_{(i-L_A),:,:} \end{bmatrix}, & i \in \{L_A+1,\dots,L_A+L_B\} \end{cases}$$

$$\boldsymbol{\mathcal{G}}_C^{(t)} \in \mathbb{R}^{(L_A+L_B)\times(R_{t-1,A}+R_{t-1,B})\times(R_{t,A}+R_{t,B})}, \quad 1 < t < T$$

$$[\boldsymbol{\mathcal{G}}_C^{(t)}]_{i,:,:} = \begin{cases} \begin{bmatrix} [\boldsymbol{\mathcal{G}}_A^{(t)}]_{i,:,:} & 0 \\ 0 & 0 \end{bmatrix}, & i \in \{1,\dots,L_A\} \\[3ex] \begin{bmatrix} 0 & 0 \\ 0 & [\boldsymbol{\mathcal{G}}_B^{(t)}]_{(i-L_A),:,:} \end{bmatrix}, & i \in \{L_A+1,\dots,L_A+L_B\} \end{cases}$$

$$\boldsymbol{\mathcal{G}}_C^{(T)} \in \mathbb{R}^{(L_A+L_B)\times(R_{t-1,A}+R_{t-1,B})\times 1}$$

$$[\boldsymbol{\mathcal{G}}_C^{(T)}]_{i,:,:} = \begin{cases} \begin{bmatrix} a[\boldsymbol{\mathcal{G}}_A^{(T)}]_{i,:,:} \\ 0 \end{bmatrix}, & i \in \{1,\dots,L_A\} \\[3ex] \begin{bmatrix} 0 \\ b[\boldsymbol{\mathcal{G}}_B^{(T)}]_{(i-L_A),:,:} \end{bmatrix}, & i \in \{L_A+1,\dots,L_A+L_B\}. \end{cases}$$

It is straightforward to verify that the network defined by these weights possesses the following property:

$$\mathbf{h}_C^{(t)} = \begin{bmatrix} \mathbf{h}_A^{(t)} \\ \mathbf{h}_B^{(t)} \end{bmatrix}, \quad 0 < t < T,$$

and

$$\mathbf{h}_C^{(T)} = a\mathbf{h}_A^{(T)} + b\mathbf{h}_B^{(T)},$$

concluding the proof. We also note that these formulas generalize the well–known formulas for addition of two tensors in the Tensor Train format (Oseledets, 2011). $\qquad\square$

**Proposition A.4.** *For any associative and commutative binary operator $\xi$, an arbitrary generalized rank 1 shallow network with $\xi$–nonlinearity can be represented in a form of generalized RNN with unit ranks ($R_1 = \cdots = R_{T-1} = 1$) and $\xi$–nonlinearity.*

**Proof.** Let $\Theta = \left(\lambda, \{\mathbf{v}^{(t)}\}_{t=1}^T\right)$ be the parameters specifying the given generalized shallow network. Then the following weight settings provide the equivalent generalized RNN (with $\mathbf{h}^{(0)}$ being the unity of the operator $\xi$).

$$\mathbf{C}^{(t)} = \left(\mathbf{v}^{(t)}\right)^\top \in \mathbb{R}^{1\times M},$$

$$\boldsymbol{\mathcal{G}}^{(t)} = 1, \quad t < T,$$

$$\boldsymbol{\mathcal{G}}^{(T)} = \lambda.$$

Indeed, in the notation defined above, hidden states of generalized RNN have the following form:

$$\mathbf{h}^{(t)} = \boldsymbol{\mathcal{G}}^{(t)} \xi \left( [\mathbf{C}^{(t)} f_\theta(\mathbf{x}^{(t)})], \mathbf{h}^{(t-1)} \right)$$

$$= \xi \left( \langle f_\theta(\mathbf{x}^{(t)}), \mathbf{v}^{(t)} \rangle, \mathbf{h}^{(t-1)} \right), \quad t = 1, \dots, T-1$$

$$\mathbf{h}^{(T)} = \lambda \xi \left( \langle f_\theta(\mathbf{x}^{(T)}), \mathbf{v}^{(T)} \rangle, \mathbf{h}^{(T-1)} \right).$$

The score function of generalized RNN is given by eq. (16):

$$\ell(X) = \mathbf{h}^{(T)} = \lambda \xi \left( \langle f_\theta(\mathbf{x}^{(T)}), \mathbf{v}^{(T)} \rangle, \mathbf{h}^{(T-1)} \right)$$

$$= \lambda \xi \left( \langle f_\theta(\mathbf{x}^{(T)}), \mathbf{v}^{(T)} \rangle, \langle f_\theta(\mathbf{x}^{(T-1)}), \mathbf{v}^{(T-1)} \rangle, \mathbf{h}^{(T-2)} \right)$$

$$\dots$$

$$= \lambda \xi \left( \langle f_\theta(\mathbf{x}^{(T)}), \mathbf{v}^{(T)} \rangle, \dots, \langle f_\theta(\mathbf{x}^{(1)}), \mathbf{v}^{(1)} \rangle \right),$$

which coincides with the score function of rank 1 shallow network defined by parameters $\Theta$.

$\square$

**Lemma 5.2.** *Let $\boldsymbol{\mathcal{E}}^{(j_1 j_2 \dots j_T)}$ be an arbitrary one–hot tensor, defined as*

$$\boldsymbol{\mathcal{E}}^{(j_1 j_2 \dots j_T)}_{i_1 i_2 \dots i_T} = \begin{cases} 1, & j_t = i_t \quad \forall t \in \{1, \dots, T\}, \\ 0, & otherwise. \end{cases}$$

*Then there exists a generalized RNN with rectifier nonlinearities such that its grid tensor satisfies*

$$\boldsymbol{\Gamma}^\ell(\mathbb{X}) = \boldsymbol{\mathcal{E}}^{(j_1 j_2 \dots j_T)}.$$

**Proof.** It is known that the statement of the lemma holds for generalized shallow networks with rectifier nonlinearities (see (Cohen & Shashua, 2016, Claim 4)). Based on Proposition A.4 and Lemma 5.1 we can conclude that it also holds for generalized RNNs with rectifier nonlinearities. $\square$

**Proposition A.5.** *Statement of Theorem 5.1 holds with $\xi(x, y) = xy$.*

**Proof.** By assumption the matrix $\mathbf{F}$ is invertible. Consider the following tensor $\widehat{\boldsymbol{\mathcal{H}}}$ :

$$\widehat{\boldsymbol{\mathcal{H}}}_{i_1 i_2 \dots i_T} = \sum_{j_1, \dots, j_T} \boldsymbol{\mathcal{H}}_{j_1, \dots, j_T} \mathbf{F}^{-1}_{j_1 i_1} \dots \mathbf{F}^{-1}_{j_T i_T},$$

and the score function in the form of eq. (2):

$$\ell(X) = \langle \widehat{\boldsymbol{\mathcal{H}}}, \boldsymbol{\Phi}(X) \rangle.$$

Note that by construction for any input assembled from the template vectors we obtain $\ell\left( (\mathbf{x}^{(i_1)}, \dots, \mathbf{x}^{(i_T)}) \right) = \boldsymbol{\mathcal{H}}_{i_1 \dots i_T}$. By taking the standard TT and CP decompositions of $\widehat{\boldsymbol{\mathcal{H}}}$ which always exist (Oseledets, 2011; Kolda & Bader, 2009), and using Lemma 3.1 and eq. (6) we conclude that universality holds. $\square$

**Theorem 5.2** (Expressivity 1)**.** *For every value of $R$ there exists a generalized RNN with ranks $\leq R$ and rectifier nonlinearity which is exponentially more efficient than shallow networks, i.e., the corresponding grid tensor may be realized only by a shallow network with rectifier nonlinearity of width at least $\frac{2}{MT} \min(M, R)^{T/2}$.*

In order to prove the theorem we will use the standard technique of matricizations. Simply put, by matricizing a tensor we reshape it into a matrix by splitting the indices of a tensor into two collections, and converting each one of them into one long index. I.e., for a tensor $\boldsymbol{\mathcal{A}}$ of order $T$ with mode sizes being $m$, we split the set $\{1, \dots, T\}$ into two non–overlapping ordered subsets $s$ and $t$, and define the matricization $\mathbf{A}^{(s,t)} \in \mathbb{R}^{M^{|s|} \times M^{|t|}}$ by simply reshaping (and possibly transposing) the tensor $\boldsymbol{\mathcal{A}}$ according to $s$ and $t$. We will consider the matricization obtained by taking $s_{odd} = (1, 3, \dots, T-1)$,

$t_{even} = (2, 4, \ldots, T)$, i.e., we split out even and odd modes. A typical application of matricization is the following: suppose that we can upper and lower bound the ordinary matrix rank of a certain matricization using the parameters specifying each of the architectures being analyzed. Then under the assumption that both architectures realize the same grid tensor (and thus ranks of the matricization coincide) we can compare the sizes of corresponding architectures. In the case of generalized shallow networks with rectifier nonlinearity we will use the following result (Cohen & Shashua, 2016, Claim 9).

**Lemma A.1.** *Let $\mathbf{\Gamma}^\ell(\mathbb{X})$ be a grid tensor generated by a generalized shallow network of rank $R$ and $\xi(x, y) = \max(x, y, 0)$. Then*

$$\text{rank}\left[\mathbf{\Gamma}^\ell(\mathbb{X})\right]^{(s_{odd}, t_{even})} \leq R\frac{TM}{2},$$

*where the ordinary matrix rank is assumed.*

This result is a generalization of a well–known property of the standard CP-decomposition (i.e. if $\xi(x, y) = xy$), which states that for a rank $R$ decomposition, the matrix rank of *every* matricization is bounded by $R$.

In order to prove Theorem 5.2 we will construct an example of a generalized RNN with exponentially large matrix rank of the matricization of grid tensor, from which and Lemma A.1 the statement of the theorem will follow.

**Lemma A.2.** *Without loss of generality assume that $\mathbf{x}_i = \mathbf{e}_i$ (which can be achieved since $\mathbf{F}$ is invertible). Let $\mathbf{1}^{(p,q)}$ denote the matrix of size $p \times q$ with each entry being $1$, $\mathbf{I}^{(p,q)}$ denote the matrix of size $p \times q$ with $\mathbf{I}^{(p,q)}_{ij} = \delta_{ij}$ ($\delta$ being the Kronecker symbol), and $\mathbf{b} = [1 - \min(M, R), \mathbf{0}^\top_{R-1}] \in \mathbb{R}^{1 \times R}$. Consider the following weight setting for a generalized RNN with $\xi(x, y) = \max(x, y, 0)$.*

$$\mathbf{C}^{(t)} = \begin{cases} \mathbf{1}^{M,M} - \mathbf{I}^{M,M}, & t \text{ odd}, \\ \mathbf{1}^{M+1,M} - \mathbf{I}^{M+1,M}, & t \text{ even}. \end{cases}$$

$$\mathcal{G}^{(t)} = \begin{cases} \mathbf{I}^{M,R} \in \mathbb{R}^{M \times 1 \times R}, & t \text{ odd}, \\ \begin{bmatrix} \mathbf{I}^{M,R} \\ \mathbf{b} \end{bmatrix} \in \mathbb{R}^{(M+1) \times R \times 1}, & t \text{ even}. \end{cases}$$

*Then grid tensor $\mathbf{\Gamma}^\ell(\mathbb{X})$ of this RNN satisfies*

$$\text{rank}\left[\mathbf{\Gamma}^\ell(\mathbb{X})\right]^{(s_{odd}, t_{even})} \geq \min(M, R)^{T/2},$$

*where the ordinary matrix rank is assumed.*

**Proof.** Informal description of the network defined by weights in the statement in the lemma is the following. Given some input vector $\mathbf{e}_i$ it is first transformed into its bitwise negative $\bar{\mathbf{e}}_i$, and its first $R$ components are saved into the hidden state. The next block then measures whether the first $\min(R, M)$ components of the current input coincide with the hidden state (after again taking bitwise negative). If this is the case, the hidden state is set $0$ and the process continues. Otherwise, the hidden state is set to $1$ which then flows to the output independently of the other inputs. In other words, for all the inputs of the form $X = (\mathbf{x}_{i_1}, \mathbf{x}_{i_1}, \ldots, \mathbf{x}_{i_{T/2}}, \mathbf{x}_{i_{T/2}})$ with $i_1 \leq R, \ldots, i_{T/2} \leq R$ we obtain that $\ell(X) = 0$, and in every other case $\ell(X) = 1$. Thus, we obtain that $\left[\mathbf{\Gamma}^\ell(\mathbb{X})\right]^{(s_{odd}, t_{even})}$ is a matrix with all the entries equal to 1, except for $\min(M, R)^{T/2}$ entries on the diagonal, which are equal to 0. Rank of such a matrix is $R^{T/2} + 1$ if $R < M$ and $M^{T/2}$ otherwise, and the statement of the lemma follows. $\square$

Based on these two lemmas we immediately obtain Theorem 5.2.

**Proof of Theorem 5.2.** Consider the example constructed in the proof of Lemma A.2. By Lemma A.1 the rank of the shallow network with rectifier nonlinearity which is able to represent the same grid tensor is at least $\frac{2}{TM}\min(M, R)^{T/2}$. $\square$

**Theorem 5.3** (Expressivity 2)**.** *For every value of $R$ there exists an open set (which thus has positive measure) of generalized RNNs with rectifier nonlinearity $\xi(x, y) = \max(x, y, 0)$, such that for each RNN in this open set the corresponding grid tensor can be realized by a rank 1 shallow network with rectifier nonlinearity.*

**Proof.** As before, let us denote by $\mathbf{I}^{(p,q)}$ a matrix of size $p \times q$ such that $\mathbf{I}_{ij}^{(p,q)} = \delta_{ij}$, and by $\mathbf{a}^{(p_1, p_2, \cdots p_d)}$ we denote a tensor of size $p_1 \times \cdots \times p_d$ with each entry being $a$ (sometimes we will omit the dimensions when they can be inferred from the context). Consider the following weight settings for a generalized RNN.

$$\mathbf{C}^{(t)} = \left(\mathbf{F}^{\top}\right)^{-1},$$

$$\boldsymbol{\mathcal{G}}^{(t)} = \begin{cases} \mathbf{2}^{(M,1,R)}, \ t = 1 \\ \mathbf{1}^{(M,R,R)}, \ t = 2, \ldots, T - 1 \\ \mathbf{1}^{(M,R,1)}, \ t = T \end{cases}$$

The RNN defined by these weights has the property that $\boldsymbol{\Gamma}^{\ell}(\mathbb{X})$ is a constant tensor with each entry being $2(MR)^{T-1}$, which can be trivially represented by a rank 1 generalized shallow network. We will show that this property holds under a small perturbation of $\mathbf{C}^{(t)}, \boldsymbol{\mathcal{G}}^{(t)}$ and $\mathbf{F}$. Let us denote each of these perturbation (and every tensor appearing size of which can be assumed indefinitely small) collectively by $\varepsilon$. Applying eq. (21) we obtain (with $\xi(x, y) = \max(x, y, 0)$).

$$\boldsymbol{\Gamma}^{\ell,0}(\mathbb{X}) = \mathbf{0} \in \mathbb{R}^1,$$

$$\boldsymbol{\Gamma}^{\ell,1}(\mathbb{X})_{km_1} = \sum_{i,j} \boldsymbol{\mathcal{G}}_{ijk}^{(1)} \left( (\mathbf{I}^{(M,M)} + \varepsilon) \otimes_{\xi} \mathbf{0} \right)_{im_1 j} = \mathbf{1} \otimes (\mathbf{2} + \varepsilon),$$

$$\boldsymbol{\Gamma}^{\ell,2}(\mathbb{X})_{km_1 m_2} = \sum_{i,j} \boldsymbol{\mathcal{G}}_{ijk}^{(2)} \left( (\mathbf{I}^{(M,M)} + \varepsilon) \otimes_{\xi} \boldsymbol{\Gamma}^{\ell,1}(\mathbb{X}) \right)_{im_2 j m_1} = \mathbf{1} \otimes (\mathbf{2MR} + \varepsilon) \otimes \mathbf{1},$$

$$\cdots$$

$$\boldsymbol{\Gamma}^{\ell,T}(\mathbb{X})_{km_1 m_2 \ldots m_T} = 1 \otimes (\mathbf{2}(\mathbf{MR})^{\mathbf{T-1}} + \varepsilon) \otimes \mathbf{1} \cdots \otimes \mathbf{1},$$

$$\boldsymbol{\Gamma}^{\ell}(\mathbb{X}) = \boldsymbol{\Gamma}^{\ell,T}(\mathbb{X})_{1,:,:,\ldots,:} = (\mathbf{2}(\mathbf{MR})^{\mathbf{T-1}} + \varepsilon) \otimes \mathbf{1} \cdots \otimes \mathbf{1},$$

where we have used a simple property connecting $\otimes_{\xi}$ with $\xi(x, y) = \max(x, y, 0)$ and ordinary $\otimes$: if for tensors $\boldsymbol{\mathcal{A}}$ and $\boldsymbol{\mathcal{B}}$ each entry of $\boldsymbol{\mathcal{A}}$ is greater than each entry of $\boldsymbol{\mathcal{B}}$, $\boldsymbol{\mathcal{A}} \otimes_{\xi} \boldsymbol{\mathcal{B}} = \boldsymbol{\mathcal{A}} \otimes \mathbf{1}$. The obtained grid tensors can be represented using rank 1 generalized shallow networks with the following weight settings.

$$\lambda = 1,$$

$$\mathbf{v}_t = \begin{cases} \mathbf{F}_{\varepsilon}^{-1}(\mathbf{2}(\mathbf{MR})^{\mathbf{T-1}} + \varepsilon), \ t = 1, \\ \mathbf{0}, \ t > 1, \end{cases}$$

where $\mathbf{F}_{\varepsilon}$ is the feature matrix of the corresponding perturbed network. $\square$

# B ADDITIONAL EXPERIMENTS

In this section we provide the results additional computational experiments, aimed to provide more thorough and complete analysis of generalized RNNs.

**Different $\xi$-nonlinearities** In this paper we presented theoretical analysis of rectifier nonlinearity which corresponds to $\xi(x, y) = \max(x, y, 0)$. However, there is a number of other associative binary operators $\xi$ which can be incorporated in generalized tensor networks. Strictly speaking, every one of them has to be carefully explored theoretically in order to speak about their generality and expressive power, but for now we can compare them empirically.

Table 1 shows the performance (accuracy on test data) of different nonlinearities on MNIST, CIFAR—10, and IMDB datasets for classification. Although these problems are not considered hard to solve, we see that the right choice of nonlinearity can lead to a significant boost in performance. For the experiments on the visual datasets we used $T = 16, m = 32, R = 64$ and for the experiments on the IMDB dataset we had $T = 100, m = 50, R = 50$. Parameters of all networks were optimized using Adam (learning rate $\alpha = 10^{-4}$) and batch size 250.

| $\xi(x,y)$ | $xy$ | $\max(x,y,0)$ | $\ln(e^x+e^y)$ | $x+y$ | $\sqrt{x^2+y^2}$ |
|---|---|---|---|---|---|
| MNIST | 97.39 | 97.45 | **97.68** | 96.28 | 96.44 |
| CIFAR-10 | 43.08 | 48.09 | 55.37 | **57.18** | 49.04 |
| IMDB | 83.33 | **84.35** | 82.25 | 81.28 | 79.76 |

Table 1: Performance of generalized RNN with various nonlinearities.

**Expressivity in the case of shared cores**   We repeat the expressivity experiments from Section 6 in the case of equal TT–cores ($\mathcal{G}^{(2)} = \cdots = \mathcal{G}^{(T-1)}$). We observe that similar to the case of different cores, there always exist rank 1 generalized shallow networks which realize the same score function as generalized RNN of higher rank, however, this situation seems too unlikely for big values of $R$.

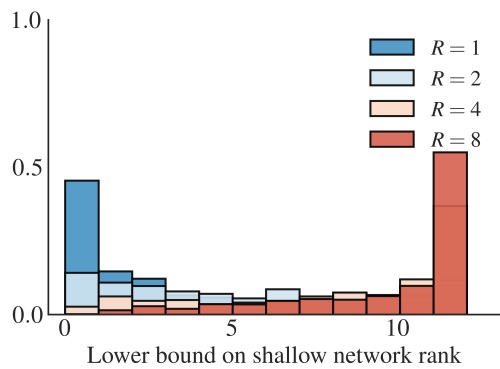

Figure 4: Distribution of lower bounds on the rank of generalized shallow networks equivalent to randomly generated generalized RNNs of ranks ($M = 6, T = 6, \xi(x,y) = \max(x,y,0)$).

Figure 5: Distribution of lower bounds on the rank of generalized shallow networks equivalent to randomly generated generalized RNNs of ranks ($M = 6, T = 6, \xi(x,y) = \sqrt{x^2+y^2}$).