# OpenReview forum: "Generalized Tensor Models for Recurrent Neural Networks"
_ICLR.cc/2019/Conference_

### Official Review · AnonReviewer1 · 2018-10-22
**Good incrementally theoretical paper with supporting experimental results. The presentation could be improved (see comments).**

**Rating:** 7
**Confidence:** 4

**Review:**

The authors extend the theoretical results of a paper previously presented in the last edition of ICLR (2018), where it was demonstrated that Recurrent Neural Network can be interpreted as a tensor network decomposition based on the Tensor-Train (TT, Oseledets et al, 2011).
While previous results covered the multiplicative nonlinearity only, the contribution of the current paper is the extension of the analysis of universality and depth efficiency (Cohen et al, 2016) to different nonlinearities, for example ReLU (Rectified Linear Unit), which is very important from the practical point of view.
The paper is well written and have a good structure. However, I found that some deep concepts are not well introduced, and maybe other more trivial results are discussed with unnecessary details. The following comments could help authors to improve the quality of presentation of their paper:
-	Section 3.1 (Score Functions and Feature Tensor) is a bit short and difficult to read.
o	Maybe, a more motivating introduction could be included in order to justify the definition of score functions (eq. 2).
o	It would be also nice to state that, according to eq. (3), the feature tensor is a rank-1 tensor.
o	I would suggest moving the definition of outer product to the Appendix, since most readers know it very well.
o	It is said that eq. 2 possesses the universal approximation property (it can approximate any function with any prescribed precision given sufficiently large M). It is not clear which is the approximation function.
-	A Connection with Tensor-Ring (TR) format, if possible, could be helpful: It is known that TR format (Zhao et al, 2016, arXiv:1606.05535), which is obtained by connecting the first and last units in a TT model, helps to alleviate the requirement of large ranks in the first and last the core tensors of a TT model reaching to a decomposition with an evenly distributed rank bounds. I think, it would be interesting to make a connection of RNN to TR because the assumption of R_i < R for all i becomes more natural. I would like to see at least some comment from the authors about the applicability of TR in the context of analysis of RNN, if possible. Maybe, also the initial hidden state defined in page 5 can be avoided if TR is used instead of TT.
-	Fig 2 shows that Test accuracy of a shallow network (CP based) is lower and increases with the number of parameters approaching to the one for RNN (TT based). It would be necessary to show the results for an extended range in the number of parameters, for example, by plotting the results up to 10^6. It is expected that, at some point, the effect of overfitting start decreasing the test accuracy.
-       When scores functions are presented (eq. 2) it is written the term "logits" between brackets. Could you please clarify why this term is introduced here? Usually, logit of a probability p is defined as L(p)=p/(1-p). What is the usage of this term in this work?
-      I think the theory is presented for a model with the two-classes only but used for multiple classes in the experimental sections. It should be necessary to make some comment about this in the paper.
-      Details about how the RNN based on TT is applied must be added. More specifically, the authors should provide answers to clarify the following questions:
(i) Are patches overlapped or non-overlapped?
(ii) What value of M is used? and is there any general rule for this choice?
(iii) How the classification in the 10-classes is obtained? Are you using a softmax function in the last layer? Are you using one weight tensor W_c per class (c=1,2,...,10). Please provide these technical details.
(iv) Please, specify which nonlinear activation sigma is used in the feature map f_\theta(x).
(v) How many feature maps are used? and, Are the matrix A and vector b learned from training dataset or only the TT-cores need to be learned?

---

> ### Author Response · Authors · 2018-11-30
> **Tensor models based on Tensor Ring, clarification on "logits" term, implementation details**
>
> Thank you for such detailed and instructive analysis of our work! We hope that following comments will help to resolve your questions.
>
> 1. The approximation function is l(X) and it approximates functions F: \mathbb{R}^{N \times T} \rightarrow \mathbb{R}. An example of such function F is a probability that given sequence X corresponds to a particular class.
>
> 2. The connection with Tensor Ring format is an appealing direction for future research. We have not investigated this connection yet, but here are some thoughts on your comments.  In particular, incorporating the circular dimensional permutation invariance property into neural networks may provide insights into how to design new types of RNNs. It would be also interesting to look at the distribution of lower bounds on the rank of TR-based tensor models (in analogy to Figure 3), as evenly distributed rank bounds may give better results than less balanced TT.
>
> 3. It is indeed the case — we will overfit if we increase the number of parameters. However, the purpose of the Figure 2 is to show that in the case of ReLU nonlinearity, TT-based generalized tensor models are more expressive than CP-based models. We believe that Figure 2 is a good illustration to this phenomena even within a current range of parameters number.
>
> 4. By “logits” we mean immediate outputs of the last hidden layer before applying nonlinearity. This term is adopted from classification tasks where neural network usually outputs “logits” and following softmax nonlinearity transforms them into valid probabilities of classes. Note that the theory supports arbitrary number of classes: we just need to approximate logit for each class with our model and then apply sigmoid nonlinearity to get a valid classifier.
>
> 5. TT-based RNN implementation details
> (i) We experimented with non-overlapping patches of size 7x7 for MNIST and of size 8x8 for CIFAR-10.
> (ii) From the analogy between vanilla RNNs and our model, M is closely related to the dimensionality of hidden state in RNN, which is typically set to tens or hundreds. In our experiments we used M=32 for visual datasets and M=50 for IMDB (as it exhibits longer temporal dependencies).
> (iii) We used the same weight tensor W for all classes and in order to get 10 probabilities we added fully-connected layer right after the output of tensor network and used softmax activation function to obtain valid probabilities.
> (iv) We used ReLU in the preprocessing feature map f_\theta(x).
> (v) Both matrix A and vector b are treated as an additional fully-connected layer and are learned together with TT-cores via backpropagation.

---

> > ### Comment · AnonReviewer1 · 2018-12-05
> > **Response**
> >
> > I thank the Authors for providing their responses to my questions.

---

### Official Review · AnonReviewer2 · 2018-11-03
**Good paper that would benefit from a meaningful and concrete toy example.**

**Rating:** 7
**Confidence:** 3

**Review:**

This paper would benefit from a meaningful and concrete toy example.

The toy example from section 3.1 eq(3) amounts to stating that the Kronecker product  of a set of eigenparts is equal to the PCA eigenpixels for  a set of complete images, or more generally that the kronecker product of a set of  image-part feature tensors is equal to the complete image-feature tensor  (assuming no pixel overlap).  Sure.  What does that buy?   Hierarchical Tucker (including Tensor Train) does indeed compute the standard Tucker mode representation of a tensor in an efficient manner using a set of sequential SVDs rather than using a single SVD per mode.   Is there anything else?  Depending on how the data is organized into a data tensor, the object representation and its properties can differ dramatically.  Section 3.1 needs further clarification.

Questions:
1. What is data tensor organization and what tensor decomposition model are you using? Tucker but implemented as a TT?
What is the resulting object representation?
2. In the context of the toy example, please give a concrete mapping between your tensor decomposition (object representation)  and RNN.

The rest of the paper is a lot of mathematical manipulation which looks correct.


Please reference the first papers to employ tensor decompositions for imaging.

M. A. O. Vasilescu, D. Terzopoulos, "Multilinear Analysis of Image Ensembles: TensorFaces,"  Proc. 7th European Conference on Computer Vision (ECCV'02), Copenhagen, Denmark, May, 2002, in Computer Vision -- ECCV 2002, Lecture Notes in Computer Science, Vol. 2350, A. Heyden et al. (Eds.), Springer-Verlag, Berlin, 2002, 447-460.

M.A.O. Vasilescu, "Multilinear Projection for Face Recognition via Canonical Decomposition ",  In Proc. Face and Gesture Conf. (FG'11), 476-483.

---

> ### Author Response · Authors · 2018-12-01
> **Data organization & object representation, mapping between RNNs and generalized tensor models**
>
> Thank you for pointing us to the relevant papers on tensor decompositions for imaging, we have already updated the related work section to include them. Also, please see the answers to the raised questions.
>
> 1. In our analysis, the data is initially represented as a matrix (collection of vectors). This representation is later translated into rank 1 canonical tensor \Phi. In its simplest form (multiplicative) the model can be written as <W, \Phi>, where W is the weight tensor. As was shown in [1], TT decomposition of W leads to the recurrent architecture. In our analysis, we move further from this representation directly to RNNs with ReLU nonlinearity, and show that this model corresponds to the ReLU-TT decomposition of the “grid tensor”, containing values of the model samples on a certain tensor product grid.
>
> 2. In order to map vanilla RNN to our model we need to redefine two things:
> 1) Trainable weights. In the case of vanilla RNN without any nonlinearity and hidden state equation h_t = A*h_{t-1} + B*x_t + c, the weights are two matrices A and B and the vector c. In the case of our model, the weights are 3-dimensional TT-cores G_t (equation 11) or a single TT-core G (equation 12) if we set all of them to be equal.
> 2) Equation for hidden state. Equations 10, 12, and 16 provide us with concrete formulas for calculating hidden states for different models. We just need to plug them in instead of corresponding formula for vanilla RNN.
> To implement our models in practice we took the original code of TensorFlow RNN Cell, redefined trainable weights, and changed the expression for hidden state computation.
>
> [1] V. Khrulkov, A. Novikov, I. Oseledets. Expressive Power of Recurrent Neural Networks. In International Conference on Learning Representations, 2018.

---

### Official Review · AnonReviewer3 · 2018-11-04
**The paper analyze connection between RNN and TT decomposition by incorporating nonlinearity. The theoretical results are very interesting while novelty is limited.**

**Rating:** 6
**Confidence:** 4

**Review:**

This paper extends the work of TT-RRN [Khrulkov et al., 2018] to further analyze the connection between RNN and TT decomposition by incorporating generalized nonlinearity, i.e., RELU, into the network architectures. Specifically, the authors theoretically study the influence of generalized nonlinearity on the expressivity power of TTD-based RNN, both theoretical result and empirical validation show that generalized TTD-based RNN is more superior to CP-based shallow network in terms of depth efficiency.
Pros:
1. This work is theoretically solid and extend the analysis of TT-RNN to the case of generalized nonlinearity, i.e. ReLU.

2. The paper is well written and organized.

Cons:
1. The contribution and novelty of this paper is incremental and somehow limited, since the analysis TT-RNN based on the product nonlinearity already exists, which make the contribution of this paper decreased.

2. The analysis is mainly on the particular case of rectifier nonlinearity. I wonder if the nonlinearities other than the RELU hold the similar properties? The proof or discussion on the general nonlinearities is missing.

Other comments:
1. The authors said that the replacement of standard outer product with its generalized version leads to the loss of conformity between tensor networks and weight tensors, the author should clarify this in a bit more details.

2. The theoretical analysis relies on grid tensor and restricts the inputs on template vectors. It is not explained why to use and how to choose the those template vectors in practice?

3. A small typo: In Figure 2, ‘m' should be ‘M'

---

> ### Author Response · Authors · 2018-11-30
> **General nonlinearities discussion, loss of conformity between tensor networks and weight tensors, template vectors**
>
> Thank you for your comments! Please, see the answers to your questions below.
>
> 1. The detailed proof we provide in the paper indeed refers to the ReLU nonlinearity only. Due to the constructive nature of our proof, it is not easily generalized to the arbitrary associative and commutative binary operator, and we highly doubt that it will work in general. However, even without solid theoretical justification, we can implement generalized tensor networks with various nonlinearities and compare them empirically, which we do in Section B of the appendix. As we can see, the right choice of nonlinearity for particular dataset may lead to boost in the performance and it will be interesting direction of research to analyze them more rigorously from both theoretical and practical viewpoints.
>
> 2. Before introducing the concept of generalized tensor networks, we had full correspondence between the score functions of scalar product form (equation 2) and the score functions of tensor decomposition form (equations 6 and 8). It ensures that tensor networks are universal function approximators and allows us to focus on another important properties, such as expressivity. However, after replacing the outer product with different operator and declaring the expressions from equations 14 and 16 to be the score functions, we can no longer state that they can be represented in a form of equation 2. Specifically, we can no longer guarantee the existence of corresponding weight tensor W (and, thus, universality), the existence of which was trivial in the case of standard tensor networks with multiplicative nonlinearity.
>
> 3. The use of template vectors is motivated by the discussion in [1]. In order to be able to achieve zero classification error for the model under analysis, the data has to satisfy two assumptions: label has to be completely determined by the instance, and the input vectors may be quantized into one of the M templates. The assumption that natural images possess these properties is based on various empirical studies. For example, it was shown in [2] that small image patches of sizes 2x2, 4x4, 8x8, 16x16, and so on, can be effectively modeled by a GMM of size 64. We believe that similar properties also hold for sequential data appearing in NLP tasks, however, this assumption requires further investigation.
>
> [1] N. Cohen, O. Sharir, A. Shashua. On the expressive Power of Deep Learning: A Tensor Analysis. In Conference on Learning Theory, pp. 698 - 728, 2016.
> [2] D. Zoran, Y. Weiss. Natural images, Gaussian Mixtures and Dead Leaves. In Advances in Neural Information Processing Systems, pp. 1745 - 1753, 2012.

---

> > ### Comment · AnonReviewer3 · 2018-12-05
> > **response**
> >
> > Thank you for your response. I believe the paper could be improved and would be more interesting if some analysis on general nonlinearities is provided.

---

### Meta-Review · Area_Chair1 · 2018-12-15
**Solid work + references should be extended.**

**Confidence:** 5
**Recommendation:** Accept (Poster)

**Metareview:**

AR1 finds that extension of the previously presented ICLR'18 paper are interesting and sufficient due to the provided analysis of universality and depth efficiency. AR2 is concerned with the lack of any concrete toy example between the proposed architecture and RNNs. Kindly make an effort to add such a basic step-by-step illustration for a simple chosen architecture e.g. in the supplementary material. AR3 is the most critical (the analysis TT-RNN based on the product non-linearity done before, particular case of rectifier non-linearity is used, etc.)

Despite the authors cannot guarantee the existence of corresponding weight tensor W in less trivial cases, the overall analysis is very interesting and it is the starting point for further modeling. Thus, AC advocates acceptance of this paper. The review scores do not indicate this can be an oral paper, e.g. it currently is unlikely to be in top few percent of accepted papers. Nonetheless, this is a valuable and solid work.

Moreover, for the camera-ready paper, kindly refresh your list of citations as a mere 1 page of citations feels rather too conservative. This makes the background of the paper and related work obscure to average reader unfamiliar with this topic, tensors, tensor outer products etc. There are numerous works on tensor decompositions that can be acknowledged:
- Multilinear Analysis of Image Ensembles: TensorFaces by Vasilescou et al.
- Multilinear Projection for Face Recognition via Canonical Decomposition by Vasilescou et al.
- Tensor decompositions for learning latent variable models by Anandkumar et al.
- Fast and guaranteed tensor decomposition via sketching by Anandkumar et al.

One good example of the use of the outer product (sums over rank one outer products of higher-order) is paper from 2013. They perform higher-order pooling on encoded feature vectors (although this seems to be the shallow setting) similar to Eq. 2 and 3 (this submission):
- Higher-order occurrence pooling on mid-and low-level features: Visual concept detection by Koniusz et al. (e.g. equations equations 49 and 50 or 1, 16 and 17 realize Eq. 3 and 13 in this submission)
- Higher-Order Occurrence Pooling for Bags-of-Words: Visual Concept Detection (similar follow-up work)

Other related papers include:
- Long-term Forecasting using Tensor-Train RNNs by Anandkumar et al.
- Tensor Regression Networks with various Low-Rank Tensor Approximations by Cao et al.

Of course, the authors are encouraged to cite even more related works.